# "I just felt supported": Transgender and non-binary patient perspectives on receiving transition-related healthcare in family planning clinics

Natalie Ingraham[1]*, Lindsey Fox[2], Andres Leon Gonzalez[3], Aerin Riegelsberger[3]

**1** Department of Sociology, California State University, Hayward, CA, United States of America, **2** Osher Center for Integrative Medicine, University of California San Francisco, San Francisco, CA, United States of America, **3** Department of Social Work, California State University, Hayward, CA, United States of America

* natalie.ingraham@csueastbay.edu

## Abstract

Transgender and non-binary people face challenges in accessing gender affirming hormone therapy. Family planning clinics across the United States have greatly expanded transgender care services in the last ten years offering increased access to these services. This national qualitative study describes transgender and non-binary patients' experiences of receiving transgender care in family planning clinics. We completed 34 in-depth interviews with transgender and non-binary people over age 18 who had received transition-related care at a family planning clinic in the last year from 2019–2020. We analyzed interview data in Dedoose using constant comparative analysis and inductive thematic analysis. Patients reported overwhelmingly positive experiences at family planning clinics and were especially surprised at the ease and speed of the informed consent process. Barriers to care remain for patients in rural areas, low income patients, and patients who need specialized care. Some of the barriers relate to the gender binary and transphobia built into the medical systems, which cause patients and providers to have to find "work arounds" the binary medical and insurance systems. Patients also shared their idealized visions of transition related care that center on strong referral networks and hiring of LGBTQ staff at the clinics. Family planning clinics currently provide affirming and supportive care, especially those that use the informed consent model. Family planning clinics could provide increased access to transgender healthcare outside of major metropolitan areas and for transgender and gender nonconforming clients across the lifespan.

## Introduction

Transgender and/or non-binary people face extensive barriers accessing transition-related care in the United States. Transition-related care in this context refers to gender affirming care to support medical transition, primarily hormone therapy (HT). This can be due to interpersonal stigma enacted in healthcare provider microaggressions [1, 2], invasive questions from

**Data Availability Statement:** Data cannot be shared publicly because of the small target population and possible identification of participants. If required, de-identified data may be

available for researchers who meet the criteria for access to confidential data. Data can be requested from the CSUEB IRB by emailing irb@csueastbay. edu.

**Funding:** NI received an CSU East Bay internal grant (Faculty Support Grant) for this project. There are no grant numbers. The sponsors did not play any role in study design, data collection, analysis, decision to publish or preparation of the manuscript.

**Competing interests:** The authors have declared that no competing interests exist.

providers about their gender unrelated to the reason for seeking care [3, 4], or verbal or physical assault in healthcare settings [5]. They also face structural barriers to transition related care, including a lack of insurance coverage of transition-related medications or procedures [6], refusal to provide transition-related care from religious healthcare systems [7], or an inability to find providers with appropriate training due to a lack of widespread medical education on transgender health in US medical schools [8].

A previous study by the same research team [9] found that family planning clinics represent a promising increased access to transition-related care given their widespread geographical access and mission alignments with providing care to minoritized populations. Although several studies have examined transgender reproductive healthcare needs and experiences [10, 11] and transgender individuals navigation of primary healthcare settings [12], the authors are not aware of any studies examining transgender and non-binary patients' experiences with transition-related care in family planning clinics.

Building on research on stigmatized healthcare for transgender individuals [2] and transgender healthcare access and experiences as noted above, this study examines the increase in transition-related care in abortion clinics in the United States from the patient perspective. We ask what are transgender or non-binary people's experiences with receiving transition-related health care at family planning clinics?

## Methods

This qualitative study expands on a pilot project focused on the provider perspective of transgender care in family planning clinics. This arm of the study focused on patient perspectives of receiving care in family planning clinics using inductive thematic analysis with elements of grounded theory methodology [13], such as constant comparative analysis of data during and after data collection.

### Ethics statement

The Institutional Review Board at California State University, East Bay provided ethical approval for this study. Verbal consent was obtained after reviewing the consent form sent to participants via email and recorded as part of the transcript for documentation. Participants were asked to choose or were assigned a pseudonym for study purposes; these are used in the results section below.

### Recruitment & sampling

Participants were primarily recruited through clinics that previously participated in another arm of the study by the primary investigator focused on staff perspectives of transgender care in family planning clinics. 82 clinics whose websites stated they offered transgender care were contacted for passive recruitment assistance (posting flyers) and 26 (31%) agree to assist with recruitment. Interview participants were chosen using purposive and snowball sampling. Purposive sampling interviews focused on ensuring diversity in participants' gender identity and race as well as location of the clinics where they received care. Snowball sampling was conducted by asking the interview participants to recommend other transgender and non-binary patients that might qualify. Several transgender social media outlets also posted study recruitment flyers.

### Procedures

Participants contact the primary investigator directly (through text, phone or e-mail) after seeing flyers in their clinics or being referred by another participant. Participants were screened

for inclusion criteria: over age 18, identify as transgender or non-binary, and have received transition-related healthcare at a family planning clinic in the last year. Participants were asked for the location of their clinic and their self-identified racial/ethnic identities to ensure diversity in the geographic regions and racial makeup of participants. Interviews were tape recorded and transcribed after informed consent was obtained. Interviews lasted between 15–45 minutes and were conducted by phone or video conference, according to participant preference. Participants were compensated $20 per person for participation in the study (given via electronic gift card through email).

## Research team

The interviews were conducted by all four team members (PI and three research assistants). The PI conducted 16 of the 34 interviews. She is a White cisgender queer woman who conducted this study and supervised three student research assistants (two graduate students and one undergraduate student) as an assistant professor. All three research assistants identify as members of the LGBTQ community: one is a Chicano transmasculine queer man (n = 4 interviews), one is a White, transmasculine and non-binary queer person (n = 10 interviews) and one is a White, cisgender queer woman (n = 4 interviews). As an ally of the transgender community, the PI has an investment in the potential policy and clinical implications of the findings of this study.

## Analysis

Qualitative analysis of in-depth interviews was completed by the entire research team using Dedoose, an online mixed methods software that allows for collaborative work. We began open coding by reviewing interview field notes and transcripts to create the initial code book. This first round of codes were combined into categories using the constant comparative process [14] alongside regular memoing to document code and theory development. As new codes and theoretical concerns emerged, previous participants were re-contacted (n = 10) to ask about these topics, although we did not pursue theoretical recruitment based on codes as in traditional grounded theory due to limited recruitment avenues.

## Results

Participants represent 12 clinics across 8 states and a mixture of independent clinics and clinics that are part of a larger network (Table 1). Participants range in age from 18–66 (mean 27.65, 9.13 SD, median 26). Twenty-three (23) of the 34 were White only (67%), 6 (18%) were biracial or multiracial (3 of these 6 were Hispanic/Latinx), 2 (6%) identified as only Hispanic/Latinx and the remaining 3 (9%) were Black, Middle Eastern, or East Asian only. Seventeen (50%) participants were transgender men, 13 (38%) were transgender women and 4 (12%) were non-binary or agender people. Many participants had only visited the family planning clinic where they were recruited for one visit, though several had been going to the same clinic for 6 months or longer. Participants primarily sought medication transition resources at the family planning clinics, including hormone therapy (HT) or gender affirming hormones, though a few also had sexually transmitted infection testing or other screening services performed such as pap smears and breast/chest exams. The following sections detail participants' experiences with transition-related care at family planning clinics, highlighting their overall positive experiences, surprise at the informed consent process, remaining access barriers to care, finding "work arounds" the gender binary medical system, and idealized visions of transition related care.

**Table 1. Participant demographics (n = 34).**

| Participant Characteristic | N(%) |
|---|---|
| Gender | |
| Transgender woman | 13 (38%) |
| Transgender man | 17 (50%) |
| Non-binary or agender | 4 (12%) |
| Race/Ethnicity | |
| Caucasian only | 23 (67%) |
| Hispanic/Latinx only | 2 (6%) |
| African American only | 1 (3%) |
| East Asian only | 1 (3%) |
| Middle Eastern only | 1 (3%) |
| Biracial | 6 (18%) |
| Age | |
| Average (SD) | 27 (9.13) |
| Median | 26 |
| Range | 18–66 |
| Type of Clinic | |
| Independent Clinic | 13 (38%) |
| Networked Clinic | 21 (62%) |

## "Worth the (short) wait"—Positive patient experiences

Overall, participants have had generally positive experiences with providers, staff members, and accessing care at family planning clinics. The visit process was fairly smooth for most participants, with just a few reporting long wait times. Most said the wait times were fairly short compared to past experiences at other clinics. Participants who noticed the intake forms reported positive reactions and satisfaction with the answer options available to them to accurate describe their legal and preferred or chosen names, pronouns, sex assigned at birth, and gender identity. Derek (trans man, 27, Black) noted that he was "worried about the whole pronoun thing, how I would be received. Everyone seemed to be well educated and well versed as far as asking me what I wanted to be called, using my preferred name because during that time I didn't have my name legally changed. They made sure to make me feel comfortable. That's what really stands out actually now that I'm thinking of, even the [city name] location, everyone's just really nice."Derek's experience of staff providing comfort and knowledge around correct name and pronoun usage was common among many participants. Participants noticed that the staff had taken time to do research and learn new terminology and skills in response to patients' needs and shifting language around gender identity. Bobby (trans man, 28, Middle Eastern) described this as feeling like the staff were very "progressive" and willing to seek out answers if they don't know information.

Provider and staff interaction is where family planning clinics really shone for participants. They overwhelmingly reported respectful staff, from the front desk to security guards, and knowledgeable providers. Participants felt supported by staff members during their visit(s). Many participants felt supported in their first, and for many, only visit to the clinic to initiate HT. The participants that had made several visits to the same clinic, usually over 3–6 months, reported consistent positive experiences at each visit. None of our participants reported experiences that were negative enough to decline follow up visits, if needed.

Staff support extends to focusing on patient agency in decision-making, especially as a key aspect of the informed consent process. Amelia (trans woman, 28, White) described the

informed consent process as one that gave her many options, but allowed her to make all final decisions:

> Throughout that whole [informed consent] process it was kind of presented to me as these are your options. These are your options of what you can do and we will help you with that rather than, here are the things that we are going to do. I think having that option, and having that as a conversation rather than, again, feeling like you're trying to prove something to someone, I think that was huge.

As described earlier, patients loved the speed of the informed consent process, but the staff also prioritized patient understanding and agency as well. Alex (non-binary, 22, biracial) was one of many participants who was shocked at the speed of the informed consent process and described it as "almost too easy":

> I went in. Then they asked me questions. Then they wrote me a prescription and then that's it. It was almost too easy. I feel like when people think about [HT], you think about it like, "Oh, my god, it's so difficult." There are a lot of barriers, but maybe because I'm in a more independent place in my life I don't know I can just go and do it. I never thought of it as being easy before. I don't know. I thought of it as more inaccessible in my mind."

This shift from provider as keeper of all knowledge and final decision maker seems to shift during the informed consent process for HT with particular focus on patient goals or patients having to "prove" something about their gender identity or experience to qualify for hormones, as they may have had to in the past when mental health letters were required.

## The surprisingly quick informed consent process

Almost all patients interviewed were seen at clinics that used an informed consent model for starting hormones, meaning they do not require any referrals or letters from mental health providers before hormone therapy can begin. They do an extensive informed consent process with the patient, explaining the various medications and their risks and benefits. Patients often got their prescription for the desired hormones at their first visit, if not their second. The only reported delays for this to a 2nd or 3rd visit were for patients who needed to wait until their bloodwork came back before getting the prescription.

Patients had usually done extensive research on HT before arriving at their appointments and were often shocked a letter from a mental health provider was not required due to the informed consent process. The assumption that clinic protocols may require a letter from a mental health provider represents a potential barrier for transgender and non-binary people seeking HT. Shifting to the informed consent model allows for an "almost too easy" process that brought participants relief after their initial clinic visits. Participants were often surprised at the speed of this process, with many expecting it to take much longer to get their prescription. While the wait times varied, most participants were able to get appointments within the same week they called to schedule them and get their prescription at that first visit.

## Remaining barriers to care access

Despite their positive experiences with clinic staff and the informed consent process, barriers to care remain for transgender and non-binary patients in the study. Remaining barriers included lack of financial coverage of visit and prescription costs, distance to clinics, limitations of student health centers, and Catholic provider restrictions.

**Financial limitations.** Several participants reported paying out of pocket for services at their clinic because they did not currently have insurance coverage or their insurance coverage did not cover their services. Participants paying out of pocket discussed the budgeting required to be sure they could cover the visit and the cost of the prescriptions. Ty (trans man, 21 White) attributed these concerns to the American medical model and noted, "It sometimes feels very scary just going into a doctor's office and not knowing who you're going to see, or not knowing how much it's going to cost. And like, it feels very scary and very predatory sometimes." This fear of unknown costs of treatment was shared by many participants, several of whom did not have insurance coverage at the time of the interview.

Participants without insurance coverage reported that clinics and pharmacies offered them medication discount cards that helped subsidize the cost. Derek also noted that the family planning clinic he visited came "highly recommended" for people who needed to pay out of pocket due to lack of insurance coverage. These issues are reflective of larger concerns indicating that trans and non-binary people are more likely to be low income than their cisgender counterparts [15] making affordability of medical care, and the cost of transportation to that care, critically important.

**Geographic distance from clinics.** Distance was perhaps the most common complaint owing to many participants living in more rural areas. These participants reported having to drive over an hour to reach even a local family planning clinic in their area that provided trans care. S (trans man, 26, White) noted that most family planning clinics are in large urban cities, and he said he felt "lucky" there was one in his. He continued,

> "I don't have to travel far for that, but even transportation for certain folks is not always accessible. If there is someone up in the middle of nowhere [more rural state], which is very likely, who is trans, and they can't get to a nearby [family planning clinic], and they don't know if they feel safe going to their primary care provider, that breaks my heart. That shouldn't be the case anywhere. We should just be able to know that if I'm going to my healthcare provider, it's not—it should be a non-issue."

The barrier of geographic access for transgender and/or non-binary people in more rural areas contributes to the lack of access to care overall, since many primary care providers may not be trained in gender affirming care provision.

**Student health care center knowledge gaps.** While less than 5 of our participants were currently full-time students (undergraduate and graduate), they did highlight a lack of knowledge and access via their student health centers. Amelia (trans woman, 28, White) described her experiences seeking care as a graduate student during her early transition. She said that the staff were "supportive and friendly" but didn't have the resources or training to help her start her medical transition. She said, "I was not their first case but I think one of their first like, somebody coming out and relying on them because I was a grad student so that's where my services were penned in just because of how the insurance worked. I had to go there first." She points out an important issue that many full time students rely on university health care centers for their primary care based on required student health insurance coverage. Two participants were still covered by their student health insurance and were able to get insurance coverage for their transition-related care, but one reported challenges getting their student health insurance to cover the outside services from the clinic. One student participant joined with the campus diversity office to push back against policy changes that banned coverage of trans care in their conservative state.

**Catholic-affiliated providers.** Finally, an emerging issue that we found among a few participants is the role that Catholic affiliated providers may have on participants' previous

experiences of seeking transition-related care before arriving at family planning clinics. While this was not a wide-spread issue among participants, several reported being turned away or referred out from Catholic providers who were unable to give them transition related care due to policies banning this type of care. Gwen (trans woman, 35, White) described how her provider was limited in the types of care they could offer due to being part of a Catholic health network. She said, "When I first went to my primary care physician, and then she saw the medications that I was on and asked about them, I said I was transitioning. She actually was very affirming, but the answer was, well, I have other people who have asked me about care and I can't provide it because we're in a Catholic health network." Although none of the three participants who sought care at Catholic facilities reported mistreatment at their facilities due to their gender identity, they were also not able to be served there, which limits their access to medical care more broadly.

## Finding "work arounds" the binary medical system

Participants reported that the family planning clinics were good at finding "work arounds" to problem solve issues that came up during their visits or during the process of getting their HT prescriptions. These issues seem to mainly revolve around the gender binary built into the medical system from insurance companies to pharmacies, to billing codes.

Many participants had not yet changed their legal gender marker or their legal names, so there were often differences between the name they used at the clinic (their chosen name) and the name on their insurance cards and identification materials. Hannah (trans woman, 21, White) described having to provide additional documentation of her legal name change as a clear issue, since she had to see her deadname–her previous legal name–on the clinic's patient portal until the documentation was updated Patients reported that clinics would bill insurance companies for covered services according to assigned sex at birth e.g. cervical screenings for transgender men and including HT lab costs with those claims rather than submitting the labs separately. This resulted in billing challenges for participants at times. Avery (trans woman, 23, biracial) described the additional work needed to address what was often flagged as an error by the insurance company: "So, what we had to do is, I had to tell them—I had to get on the phone with them and explain that to them, and then, my provider went through then, they coded it differently. Instead of doing gender identity disorder, they did it under some other code, so that the insurance would cover it and that's how, ultimately, I got it paid for." These work arounds were required for several types of procedures and represent additional labor of both the patient and provider in order to get services paid for by insurance companies.

Another work around that needed to be done by clinics was addressing medication shortages, primarily for participants using feminizing hormones like estrogen/estradiol. Patients reported delays in their ability to fill prescriptions, like Amelia (trans woman, 28, White) who found that her initial order for injection estrogen was on a three month back order. When she contacted the clinic, they had it "worked out within 24 hours" to switch her to a tablet version of estrogen instead. This made her feel that the staff and providers there "had her back" because they were willing to problem solve for situations out of their control.

## Idealized visions of transition care

Participants were all asked what advice they would give other family planning clinics considering adding transgender care. Many were unsure or didn't have any specific suggestions, but those that did centered on improved specific knowledge of hormone dosage, legal document assistance, and increased numbers of visibly LGBTQ staff members, especially transgender and non-binary people. Gwen (trans woman, 35, White) summarized the most important part

of idealized transition care as the clinic staff attitudes and training. She said, "I think the biggest thing is to show that you're not only willing to listen to trans people who are coming in seeking care, but that you've already done a lot of the legwork, that you've talked to places that are already doing this work. Not trying to just go off the barest standards, but are actively trying to keep up with research and understand when new evidence arrives that might need to change the standard of care. [. . .] Trying to understand the diversity of the population who needs this care." For her, and many others, it was not enough for staff to only know the basics of transgender care.

Several participants wanted access to providers with more specific knowledge, ideally access to an endocrinologist specializing in cross-sex hormones who could adjust dosages more frequently or address potential co-occurring conditions that could impact hormone use or vice versa. Another participant, Jeong (trans woman, 25, East Asian) wanted her clinic to provide access or direct referral to an endocrinologist, also for dosage adjustments. AJ (non-binary, 26, White) also wanted more HT experts who could provide insight on adjusting medication dosages, especially "with non-binary people who don't necessarily want to become or transition as a man or a woman, you know." While family planning clinic staff were trained in basic HT protocols, patients found that referrals to endocrinologists may be needed for more complicated cases.

Participants were mixed on preferences for the structure of clinics for transition-related care. Some preferred a structure like a primary care practice or HMO where all of their health needs could be addressed in one place with providers who could provide more in-depth explanations about the medical processes involved in transition. Nick (trans man, 22, White) described the need for additional information as well, noted that he wanted "someone who'll actually explain, someone who has the knowledge to actually explain what's happening well and then who can give support and more than just, "Here's a prescription." [. . .] And I feel like I should be able to talk about not just transitioning but my other health needs and how those affect it. [. . .] So I would like it to be like a [primary care provider], like a regular clinic setting. I would also like someplace that is LGBT friendly specifically." Others would prefer to have more LGBTQ clinic access outside of major metropolitan areas.

Participants said their ideal clinics would provide assistance with legal name changes or gender markers. Ty (trans man, 21, White) commented, "I think that would be amazing [to have help with changing my name] because I haven't really begun that process, but it is definitely intimidating. Any kind of release and paperwork. I think a lot of people could benefit from like having someone to help them with that." Several clinics in our sample do provide these services, and participants reported relief and happiness that they could stay "in house" for the trans medical care and these types of support services. Clinics should also hire transgender and non-binary staff members that can help guide participants through legal changes or the medical processes. Both Bobby (trans man, 28, Middle Eastern) and Derek (trans man, 27, Black) said that clinics with visibly queer and/or transgender staff members made them more comfortable. They both described trusting queer and/or transgender staff members more due to their lived experiences as part of their community. Bobby elaborated, "You need to employ trans people to help. Because you can read in a book all day, but until you are something you're not. . . You can understand to an extent, but you don't understand." Participants felt strongly that hiring transgender staff helped them feel more welcomed and safer in family planning clinics where that was present.

## Discussion

This study of 34 transgender and/or non-binary patients from family planning clinics found that, overall, patients found the clinics to be well-equipped to handle basic transition-related care and treat transgender and non-binary clients with culturally competent respect and kindness. These positive experiences, especially for participants who had utilized family planning clinics services for several months, indicate that this could results in less delays in care or reluctance to seek care at all. This is important since these delays or refusal to seek care based on negative provider experiences remain a key issue in transgender health care [5, 12].

The struggles with the gender binary inherent in medical and insurance systems remains and participants report that many clinics have already found policies and practices that work around this binary to provide quality care for transgender and/or non-binary patients. Gender binary language may also make non-binary patients who do not identify as transgender more marginalized. One possible consideration for terminology expansion in this field is the concept of gender modality [16], which allows for a broader conception of the differences between gender assigned at birth and current gender identity.

This study is the one of the few studies to qualitatively explore transgender and non-binary people's experiences receiving transition-related care at family planning clinics. Patients generally positive experiences support the argument that family planning clinics expansion into coverage of transition related care could pose a unique opportunity for increased access to this care outside of major metropolitan areas that have LGBTQ or transgender health clinics. We found that participants perceive provider uncertainty about more complicated HT medications or impacts on other existing conditions. Stigma generated by this uncertainty and lack of knowledge can mean that patients are reluctant to ask more detailed questions or push providers for more updated clinical information due to fears of not being able to access HT or transition related care at all.

### Limitations

As with all studies that rely on convenience sampling, this study may not accurately represent the experiences of all transgender and non-binary people who get care at family planning clinics. We did engage in purposive sampling to ensure diversity in participant gender identity and racial diversity in order to better address intersections between race and gender identity that impact care [17]. We were also limited by the reality of immense time and political pressure on family planning clinics in the current climate, where many clinics simply did not have the bandwidth to participate in aiding with recruitment of participants, which limited our geographic spread across the United States.

### Conclusions

We were heartened to see that, unlike much previous qualitative research on non-binary patient experiences with medical care and transition-related care specifically, our participants report overwhelmingly positive experiences in family planning clinics. We argue that family planning clinics should strongly consider adding transition related care to their existing practices to continue to expand access to this care, especially for transgender and non-binary people in more rural areas and those without access to private transportation.

Clinics should work closely with transgender and/or non-binary communities in their area as they add this care and note that it was particularly meaningful for participants to see visible queer and transgender staff members working at these clinics and created an immediate sense of safety. Family planning clinics should also consider building strong referral networks to local endocrinologists or other HT-knowledgeable practices who can care for clients who may

need more than the starting, standard HT protocols available in most family planning clinics. Future research should continue to explore patient perspectives of transition-related care, particular for non-binary people and transgender and non-binary people of color and/or transgender and non-binary people with disabilities whose intersectional identities may impact transition-related care in unique ways.

## Acknowledgments

We would like to thank the participants for sharing their time and stories for this study, as well as our family planning clinic partners for their assistance with recruitment.

## Author Contributions

**Conceptualization:** Natalie Ingraham.

**Data curation:** Natalie Ingraham, Lindsey Fox, Andres Leon Gonzalez, Aerin Riegelsberger.

**Formal analysis:** Natalie Ingraham, Lindsey Fox, Andres Leon Gonzalez, Aerin Riegelsberger.

**Funding acquisition:** Natalie Ingraham.

**Methodology:** Natalie Ingraham.

**Project administration:** Natalie Ingraham.

**Supervision:** Natalie Ingraham.

**Writing – original draft:** Natalie Ingraham, Lindsey Fox, Andres Leon Gonzalez, Aerin Riegelsberger.

**Writing – review & editing:** Natalie Ingraham.

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
