## [Decision Letter · Decision Letter 0]

5 Apr 2022

PONE-D-22-01966“I just felt supported”: Transgender and Non-Binary Patient Perspectives on Receiving Transition-Related Healthcare in Family Planning ClinicsPLOS ONE

Dear Dr. Ingraham,

Thank you for submitting your manuscript to PLOS ONE. After careful consideration, we feel that it has merit but does not fully meet PLOS ONE’s publication criteria as it currently stands. Therefore, we invite you to submit a revised version of the manuscript that addresses the points raised during the review process.

In your revised draft, please respond to each comment from each reviewer. We look forward to reviewing your updated version soon.

We look forward to receiving your revised manuscript.

Kind regards,

Amy Michelle DeBaets, PhD

Academic Editor

PLOS ONE

Journal Requirements:

2. Thank you for sending us the data set underlying the results presented in your PLOS ONE submission. We notice that some of the information included in the data set may be potentially identifying. Please ensure that the data shared are in accordance with participant consent and provide only the data that are used in this specific study. To ensure patient confidentiality, we would recommend removing the participants names from Table 2 and substitute them with a non-identifiable code. Additional guidance on preparing raw clinical data for publication can be found in our Data Policy FAQs (https://journals.plos.org/plosone/s/data-availability#loc-clinical-data).

“This study was funded by a Faculty Support Grant at California State University, East Bay.”

We note that you have provided additional information within the Funding Section that is not currently declared in your Funding Statement. Please note that funding information should not appear in the Funding section or other areas of your manuscript. We will only publish funding information present in the Funding Statement section of the online submission form.

“NI received an CSU East Bay internal grant (Faculty Support Grant) for this project. There are no grant numbers. The sponsors did not play any role in study design, data collection, analysis, decision to publish or preparation of the manuscript.”

Reviewers' comments:

Reviewer's Responses to Questions

**Comments to the Author**

1. Is the manuscript technically sound, and do the data support the conclusions?

Reviewer #1: Yes

Reviewer #2: Yes

2. Has the statistical analysis been performed appropriately and rigorously? 

Reviewer #1: N/A

Reviewer #2: N/A

3. Have the authors made all data underlying the findings in their manuscript fully available?

Reviewer #1: No

Reviewer #2: No

4. Is the manuscript presented in an intelligible fashion and written in standard English?

Reviewer #1: Yes

Reviewer #2: Yes

5. Review Comments to the Author

Reviewer #1: 1) 'Transgender and non-binary' could be read as implying that non-binary people are not trans. Is there a clearer way to state that the manuscript talks about binary and non-binary people (without implying that all non-binary people refer to themselves as trans)? One way might be to use Florence Ashley's term 'gender modality'.

2) The introduction should more clearly state what is 'transition related care' in family planning clinics. My first thought was contraception or abortion, but these arent inherently 'transition related' (though they may be gender affirming). This is explained a little in the method, but for an international readership this should be outlined at the start of the article - it would be atypical to access hormones in family planning clinics in many countries outside the US.

3) I personally would prefer to see extracts in the body of the text and discussed, rather than in a table. Having them solely in a table treats qualitative data as an abstracted entity.

4) The extracts in a table makes the results rather odd. The themes are just summaries of a rather large data set, yet despite this being very codified and reductive, no prevalence is given (ie as per a content analysis). There is a sense in which the way the results are presented doesnt do service to either the data or thematic analysis.

5) I'm really not sure this is the 'first' study (and that language is itself colonising). A simple google scholar search for 'trans fertility planning' shows a number of studies on the first page.

Reviewer #2: First, my cordial congratulations for the methodological rigor of the research and the quality of the writing of this material. This research highlights important data and information that can contribute to health planning, ensuring access to vulnerable populations, especially those in conditions or areas of lesser or difficult access.

I have small contributions, in the form of suggestions, that may contribute to a better understanding of the data presented:

1 - In line 132 the authors present: "Twenty-three (23) of the 34 were White only (67%), 7 (20%) were biracial or multiracial (non-Latinx), and the remaining 3 (9%) were Black, Middle Eastern, or East Asian only". These dice add up to 33, not 34.

2 - In line 134, the data on the profile of participants show only Latinos (N=6 or 18%). Do the authors not consider it important to characterize the others? I ask this because other research suggests that access for vulnerable people, such as transgender people, for example, is strongly associated with intersections of gender, race, class, region, etc. In the data of this research, intersectionality is evidenced and recommended, in the conclusions, as an object of further studies.

3 - Line 136 - shows data on the time the person uses or interacts with the health service. Considering the objective of the article, I suggest deepening this data, considering that positive or negative experiences in health services also result from the quality of interaction and link between people involved in providing and receiving care.

4 - Line 140 - Mention some services that people seek including "other screening services performed". My suggestion is that you can give some examples of these other experiences. This can contribute to the clarification of mistaken and reductionist ideas about the health needs of these people and practices that are limited to body transformation or sexually transmitted infections, for example.

5 - In Line 146, "Table 1: Participant Demographics (n=34)" I suggest reviewing the data: About gender, in the table an agender person is presented. My suggestion is to present this person, in the text, as part of the group of non-binary people. Also, I was in doubt about the data related to Latinos: in the text six Latin people are mentioned and in the table, two. This consideration is related to item 2 of my assessment.

Finally, I emphasize that the items indicated above, as suggestions, did not interfere with the quality of the qualitative data presented, whose results are aligned and guided by the objective of the study, thus offering support for the conclusions obtained and the recommendations offered.

Like the authors, I am also excited to access these results that demonstrate increased satisfaction and positive experiences for non-binary and transgender people seeking and using health services. Recommendations on the strengthening of care networks, including representations of trans and NB people as service providers, expansion of provision and access to care, especially for populations with less or difficult access, contribute significantly to the advancement of science.

About item 3 "Have the authors made all data underlying the findings in their manuscript fully available?" I answered NO, but I am aware and in agreement with the justification presented, by the authors, in the pdf of the manuscript.

Best regards.

6. PLOS authors have the option to publish the peer review history of their article (what does this mean?). If published, this will include your full peer review and any attached files.

Reviewer #1: No

Reviewer #2: No

---

## [Author Response · Author response to Decision Letter 0]

13 May 2022

April 12, 2022, 

PLOS One

Dear Editors and Reviewers:

Thank you for the opportunity to revise our manuscript: "’I just felt supported’: Transgender and Non-Binary Patient Perspectives on Receiving Transition-Related Healthcare in Family Planning Clinics”, which we are submitting for exclusive consideration of publication in PLOS One. 

Editorial changes: We have made the requested editorial changes to the manuscript format. We have removed the funding statement from the manuscript as well. The Methods section already contained the IRB name and consent information, but we have added this as a specific ethics statement section to clearly highlight this. This section includes the note that all names used in the manuscript are pseudonyms. 

Please see below for a detailed response to each reviewer comment:

Reviewer #1: 

1) 'Transgender and non-binary' could be read as implying that non-binary people are not trans. Is there a clearer way to state that the manuscript talks about binary and non-binary people (without implying that all non-binary people refer to themselves as trans)? One way might be to use Florence Ashley's term 'gender modality'.

Thank you for the suggestion of the Florence Ashley term. This term was new to our team. Although all participants in our study identified as transgender, including the non-binary participants, we recognize that the current language may not reflect all non-binary people’s experiences. We have changed the introduction to reflect transgender and/or non-binary people as a starting place and included a reference to the Florence Ashley pre-print piece on gender modality as suggested language moving forward in the discussion section.

2) The introduction should more clearly state what is 'transition related care' in family planning clinics. My first thought was contraception or abortion, but these arent inherently 'transition related' (though they may be gender affirming). This is explained a little in the method, but for an international readership this should be outlined at the start of the article - it would be atypical to access hormones in family planning clinics in many countries outside the US.

We have added information to the introduction to clearly state that in this context, we are referring to gender affirming care for medical transition, primarily hormone therapy.

3) I personally would prefer to see extracts in the body of the text and discussed, rather than in a table. Having them solely in a table treats qualitative data as an abstracted entity.

4) The extracts in a table makes the results rather odd. The themes are just summaries of a rather large data set, yet despite this being very codified and reductive, no prevalence is given (ie as per a content analysis). There is a sense in which the way the results are presented doesnt do service to either the data or thematic analysis.

Thank you for these two related comments. We agree. While the table formatting was an attempt to be more succinct, we recognize (as in comment 4) that this distracts from our methodological process and analysis. We have integrated participant quotes throughout the results section to reduce the abstraction and connect participant experiences to themes. 

5) I'm really not sure this is the 'first' study (and that language is itself colonising). A simple google scholar search for 'trans fertility planning' shows a number of studies on the first page.

We have amended this language to reflect that ours is one of the few studies to consider the intersection of transition-related care at family planning clinics. While it is challenging to know if you are the first (and we recognize the potential colonizing language of this), our review of the literature does show that several studies exist about transgender and/or non-binary people and family planning care (contraception, abortion), there were no other studies on those patients getting their hormone therapy in this particular clinic context. 

Reviewer #2: 

Thank you for your initial comments on the study. 

1 - In line 132 the authors present: "Twenty-three (23) of the 34 were White only (67%), 7 (20%) were biracial or multiracial (non-Latinx), and the remaining 3 (9%) were Black, Middle Eastern, or East Asian only". These dice add up to 33, not 34.

Thank you for catching this error in transcribing our participant demographics. We have updated Table 1 and the paragraph describing participants to match each other and our sample accurately.

2 - In line 134, the data on the profile of participants show only Latinos (N=6 or 18%). Do the authors not consider it important to characterize the others? I ask this because other research suggests that access for vulnerable people, such as transgender people, for example, is strongly associated with intersections of gender, race, class, region, etc. In the data of this research, intersectionality is evidenced and recommended, in the conclusions, as an object of further studies.

When updating the demographics, we also changed how we referenced participants who identified as Latinx (either only Latinx or as part of their biracial identity). 

3 - Line 136 - shows data on the time the person uses or interacts with the health service. Considering the objective of the article, I suggest deepening this data, considering that positive or negative experiences in health services also result from the quality of interaction and link between people involved in providing and receiving care.

This was an interesting comment – we had not noticed any trends or differences between participants who had only been to clinics once vs. participants who had been going for longer periods of time. We did revisit our data and add in a note in the results section under positive experiences that none of the participants would decline to return for HT services, though we did not have data about if this was only due to access vs experience with treatment. We also added a piece about participants forming relationships with providers for those who had been going to their clinic for several months. We also added a note on this to our discussion section, linking positive experiences with continuation of care vs. delays in care as noted in previous literature.

4 - Line 140 - Mention some services that people seek including "other screening services performed". My suggestion is that you can give some examples of these other experiences. This can contribute to the clarification of mistaken and reductionist ideas about the health needs of these people and practices that are limited to body transformation or sexually transmitted infections, for example.

We have added a line with examples of screening services performed – pap smears and chest exams.

5 - In Line 146, "Table 1: Participant Demographics (n=34)" I suggest reviewing the data: About gender, in the table an agender person is presented. My suggestion is to present this person, in the text, as part of the group of non-binary people. Also, I was in doubt about the data related to Latinos: in the text six Latin people are mentioned and in the table, two. This consideration is related to item 2 of my assessment.

We have amended Table 1 to combine these two categories, as suggested. We also revised the text description to clarify the number of Latinx participants.

Thank you for considering this revision of our work. Please address all correspondence concerning this manuscript to:

Natalie Ingraham, PhD, MPH

Assistant Professor

CSU East Bay, Dept. of Sociology

25800 Carlos Bee Blvd, Hayward, CA 94542

E-mail: natalie.ingraham@csueastbay.edu Phone: (510)736-3075

Sincerely,

Natalie Ingraham, PhD, MPH

Lindsey Fox, BA

Andres Leon Gonzalez, MSW

Aerin Riegelsberger, MSW

---

## [Decision Letter · Decision Letter 1]

6 Jul 2022

“I just felt supported”: Transgender and Non-Binary Patient Perspectives on Receiving Transition-Related Healthcare in Family Planning Clinics

PONE-D-22-01966R1

Dear Dr. Ingraham,

We’re pleased to inform you that your manuscript has been judged scientifically suitable for publication and will be formally accepted for publication once it meets all outstanding technical requirements.

Kind regards,

Amy Michelle DeBaets, PhD

Academic Editor

PLOS ONE

Additional Editor Comments (optional):

Reviewers' comments:

Reviewer's Responses to Questions

**Comments to the Author**

1. If the authors have adequately addressed your comments raised in a previous round of review and you feel that this manuscript is now acceptable for publication, you may indicate that here to bypass the “Comments to the Author” section, enter your conflict of interest statement in the “Confidential to Editor” section, and submit your "Accept" recommendation.

Reviewer #1: All comments have been addressed

Reviewer #2: All comments have been addressed

2. Is the manuscript technically sound, and do the data support the conclusions?

Reviewer #1: Yes

Reviewer #2: Yes

3. Has the statistical analysis been performed appropriately and rigorously? 

Reviewer #1: N/A

Reviewer #2: N/A

4. Have the authors made all data underlying the findings in their manuscript fully available?

Reviewer #1: No

Reviewer #2: Yes

5. Is the manuscript presented in an intelligible fashion and written in standard English?

Reviewer #1: Yes

Reviewer #2: Yes

6. Review Comments to the Author

Reviewer #1: (No Response)

Reviewer #2: The changes made to the text reflect the comments and contributions made in the previous revision.

The text presents clear, objective and technical language that gives greater intelligibility of the material.

Updating and changing the way in which data is presented makes it possible to establish a relationship with the discussion and conclusions, making them more understandable.

I consider that the material offers valuable contributions to the health area, which justifies my recommendation in this journal.

7. PLOS authors have the option to publish the peer review history of their article (what does this mean?). If published, this will include your full peer review and any attached files.

Reviewer #1: No

Reviewer #2: No

---

## [Editor Report · Acceptance letter]

12 Jul 2022

PONE-D-22-01966R1 

“I just felt supported”: Transgender and Non-Binary Patient Perspectives on Receiving Transition-Related Healthcare in Family Planning Clinics 

Dear Dr. Ingraham:

I'm pleased to inform you that your manuscript has been deemed suitable for publication in PLOS ONE. Congratulations! Your manuscript is now with our production department. 

Kind regards, 

on behalf of

Dr. Amy Michelle DeBaets 

Academic Editor

PLOS ONE